# Personal, Interpersonal, and Sociocultural Factors of Condom Use in Rural Indigenous Nahuas Adolescents in Mexico

**DOI:** 10.3390/children10060921

**Published:** 2023-05-24

**Authors:** Raquel A. Benavides-Torres, María de los Ángeles Meneses-Tirado, Alan Josué Ramírez-Calderón, Dora Julia Onofre-Rodríguez, Jane Dimmitt Champion

**Affiliations:** 1Facultad de Enfermería, Universidad Autónoma de Nuevo León, Ciudad Universitaria, San Nicolás de los Garza 66451, Mexico; josue.ramirezc@uanl.edu.mx (A.J.R.-C.);; 2Facultad de Enfermería, Benemérita Universidad Autónoma de Puebla, Puebla 72000, Mexico; angeles.meneses@correo.buap.mx; 3School of Nursing, University of Texas at Austin, Austin, TX 78712, USA; jdchampion@mail.nur.utexas.edu

**Keywords:** adolescent, indigenous culture, condoms, sexual behavior

## Abstract

The goal of this research was to determine the personal, interpersonal, and sociocultural predictors of condom use among rural Indigenous adolescents. Predictor variables were selected from Bandura’s Social Cognitive Theory and Leininger’s Transcultural Theory. The sample consisted of 419 Nahuas adolescents randomly selected from the total number of neighborhood blocks in a rural community in Puebla, Mexico. The instruments had acceptable psychometric characteristics (Cronbach alpha and validity scores). Multiple linear regression models were used. Results: 56.8% of participants were female, and 50.40% were students. Mean age was M = 17.5 (SD = 0.97), and the majority (63%) identified as Catholic. Age at menarche/first ejaculation (β = −1.2, *p* = 0.038), attitude toward condom use (β = 0.13, *p* < 0.001), ethnic identity (β = 0.21, *p* < 0.001), and ability to negotiate condom use (β = 0.13, *p* = 0.003) predicted (R^2^ = 22.3) condom use. This study provided a basis for integration of the cultural values of Indigenous adolescents within interventions for sexual health promotion.

## 1. Introduction

Human Immunodeficiency Virus (HIV) infection is a global health problem. Approximately 37.7 million people are living with HIV [1]. In 2022, Mexico reported 347,749 cases of HIV/AIDS (acquired immunodeficiency syndrome), with the majority of cases affecting young people and adolescents [2]. Factors such as socioeconomic status, culture, and access to medical care might increase the risk of sexually transmitted infections (STIs), such as HIV, in some indigenous communities [3].

Mexico has an Indigenous population of twelve million people, of which 10% live in the state of Puebla with the highest presence of ethnic groups [4]. Those of Nahua ethnicity constitute one of the largest Indigenous communities in México, living in areas where health, education, and economic conditions are precarious. These living conditions increase the vulnerability of adolescents who often take risks in sexual relations regardless of consequences, including STIs and unwanted pregnancies. Sexual behavior of Indigenous adolescents incorporates meanings and practices influenced by the interaction of personal biological, affective, and cognitive factors, as well as interpersonal and sociocultural environmental factors [5]. A common adolescent risk in sexual relationships is sex without condoms.

Condoms are proven to be an effective barrier method for STI prevention. Condoms protect against STIs including chlamydia, gonorrhea, syphilis, and HIV/AIDS. Factors related to condom use (CU) among Indigenous adolescents include age at menarche in females and age at first ejaculation in males; age has been related to sexual debut, multiple sexual relations, and previous STIs. In addition, sex assigned at birth, such as being a girl or a boy, impacts condom use. Women reportedly have more difficulty reaching agreement with their partners about condom use than men. They also have difficulty with correct condom use such as placement of condoms on their partners. This implies the lack of ability to negotiate CU, yet men may designate women as responsible for use of contraceptive methods [6].

Negative attitude toward condom use is another factor that may influence decision-making regarding sexual and reproductive health [7,8]. Indigenous adolescents may not use condoms in their sexual debut and in subsequent sexual relations due to negative attitudes toward condom use [9,10,11].

Ethnic identity is a factor positively linked to sexual behavior among Indigenous adolescents. Strong ethnic identity is associated with greater self-efficacy in condom use and may vary depending on gender, yet this sociocultural factor receives limited attention [12]. Indigenous adolescents during sexual initiation have little affinity for condom use that decreases within next sexual relations [10].

The Model of Safe Sexual Behavior in Nahua Adolescents (MOCSSAN) developed by Meneses-Tirado and collaborators [5] provided the theoretical basis for this study. This model describes the relationship between personal and environmental factors and how they influence safe sexual behavior among Indigenous adolescents. This model was developed deductively from Bandura’s [13] Social Cognitive Theory (SCT) and Leininger’s Transcultural Theory [14]. This model provides a broad and detailed description of contraceptive behavior among Indigenous adolescents from a social–cognitive–cultural perspective. A more detailed description of the MOCSSAN includes further explanation of the SCT and Lininger’s theoretical framework.

The SCT explains psychosocial functioning, where each person formulates their own behavior based on the interaction between internal personal factors and environmental factors. Through the triadic reciprocity model, Bandura argues that internal mental processes and the environment are the causes of behavior. Behavior is explained as the result of the continuous interaction between biological, affective, and cognitive personal factors and the environment. SCT is based on the acquisition of skills and behaviors in an operational and instrumental way, showing the individual as a proactive subject, with the capacity to encode information and carry out behaviors that have intrinsic influence as well as an influence on the context [13]. Thus, behavior is the result of the continuous interaction between personal factors (affective, cognitive, and biological) and environmental factors (interpersonal influences).

Leininger’s conceptualization of “cultural values” [14] refers to the fact that such values assign meaning to human expressions and are usually adapted in different contexts to prove the learned or transmitted significance that has an impact on health at the individual or collective level. Additionally, culture refers to individual or group ways of life that include the values, beliefs, norms, and patterns of practice shared and transmitted between generations. The author proposes through the Sunrise Model that health care providers should consider the value dimensions and fundamental norms of culture as factors that influence the life and behavior of the person.

MOCSSAN was developed based on these two theoretical frameworks and evaluated with an Indigenous adolescent population to explain safe sexual behavior, specifically condom use. The main goal of the model is to explain the relationship between personal and environmental factors, and how these may influence safe sexual behavior among adolescents from Indigenous backgrounds. Personal factors include biological (maturation of sexual organs according to sex and age), affective (attitudes), and cognitive (knowledge, self-efficacy for CU, and intentions) factors. Environmental factors include interpersonal (family, partner, and peers) and sociocultural influences (gender roles, religious beliefs, ethnic identity) resulting in safe sexual behavior (condom use).

Little research conducted in Mexico has addressed these variables leading to a paucity of information on the subject. The study of attitudes toward CU among Nahua adolescents is important for health care providers such that access to adequate sexual health counseling including family planning for the prevention of unplanned pregnancies is available for decision-making. It is important as well to provide instructions about the available measures of protection during sexual intercourse and the prevention of sexual risk behaviors for STIs including HIV/AIDS. Therefore, the goal of this study was to determine the personal, interpersonal, and sociocultural factors influencing condom use among Nahua adolescents.

## 2. Materials and Methods

### 2.1. Study Design

A descriptive, correlational, cross-sectional study conducted among Nahua adolescents of both sexes living in the rural community of Las Lomas, Puebla included 419 participants. The sample size, calculated using the statistical package nQuery Advisor, with a significance level of 0.05, a correlation coefficient of 0.10, and a power of 80%, was adequate with an estimated non-response rate of 20%.

Single-stage cluster sampling was used considering the total number of blocks in the neighborhood and the total number of adolescents per block, therefore 32 blocks were randomly selected to achieve the sample size. The inclusion criteria were: (1) adolescents between 15 and 19 years of age, (2) sexually active, (3) not living with a sexual partner at the time of the interview, and (4) self-identified as Indigenous or speaking an Indigenous language (IL) or had at least one of the parents speaking an IL. Spanish proficiency was not an inclusion criterion.

### 2.2. Measurements

A socio-demographic questionnaire included information on participant age, schooling, sex, and age at menarche/first ejaculation. Additionally, five measures were used to assess personal factors. The subscale of hedonistic beliefs about condom use [15] was used, holding seven items with 5 Likert-type response options: 1 (completely disagree) to 5 (completely agree). The highest score corresponded to positive attitudes about condom use. The STI, AIDS, and pregnancy knowledge questionnaire with twenty-four questions [15] was also used. The response options are true, false, and do not know. Higher scores indicated greater knowledge.

Personal cognitive factors such as self-efficacy and the ability to negotiate condom use were measured through two subscales from the Brafford and Beck instrument [16]. The condom placement self-efficacy subscale contains twelve questions and the assertiveness in condom use subscale with seven statements, with five response options on a Likert-type scale: 4 (completely agree) to 0 (completely disagree); the higher the score, the greater the person’s self-efficacy and ability to negotiate condom use. Intention to use condoms was assessed with four questions measuring the probability of using condoms in the next three months [17]. Five response options for this scale ranged from 1, very unlikely, to 5, very likely.

Interpersonal factors were measured using three instruments. The sexual communication scale was used to measure communication between parents and children. It has five questions with options ranging from 1 (not at all) to 5 (very much) [18]. Another subscale from the Brafford and Beck instrument mentioned above was used to measure the partner approval of condom use through four questions. The peer norms scale assesses perceived popularity among friends for sexual behavior [19]. It has five statements with response choices ranging from 0 none to 3 all. A low score indicated absence of popularity.

Sociocultural factors were measured using three scales. The gender role in sexual relations scale assesses the frequency of gender-equitable attitudes toward partners that influence sexual relations. It has twenty-four items with options from 1 (agree) to 3 (disagree) [20]. The religious belief scale evaluates the importance of rituals and spirituality in people’s lives. It has 19 items with response options ranging from 1 (totally disagree) to 5 (totally agree). High scores mean favorable attitudes toward religion [21].

The ethnic identity scale [22] was applied to measure sociocultural factors; it has 17 statements that measure the variable ethnic identity, with response options on a Likert scale from 1 (does not describe me at all) to 4 (describes me very well). The overall score ranged from 17 to 68, so the higher the score, the higher the degree of ethnic identity. Otherwise, it shows a diffused identity with low interest and commitment to the ethnic group.

Regarding condom use, the safe sexual behavior questionnaire [23] was applied. The subscale called condom use was applied (items 1–7). The responses were presented with four options on a Likert-type scale: 1 (never), 2 (sometimes), 3 (almost always), and 4 (always). The higher the score, the greater the frequency of condom use.

### 2.3. Legal and Ethical Considerations

The study adhered to the guidelines of the Regulations of the General Health Law on Health Research in Mexico [24] and was approved by the Research Ethics Committee of the School of Nursing of the Autonomous University of Nuevo Leon (FAEN-UANL). An official letter issued by the FAEN was delivered to the Rural Health Center in which the census of adolescents and the map of the community were requested to determine the total number of blocks.

### 2.4. Study Procedures and Data Collection

Data collection was carried out once the randomly selected blocks were identified on a map. The area was visited house by house in the company of a community representative. Once the adolescent was located, the inclusion criteria were evaluated in private. Data collection was conducted over a maximum of three visits to the participants’ homes, with an explanation to the parent/guardian as well as to the adolescent(s) of the importance and goals of the study.

Subsequently, the parent/guardian was asked for informed consent to authorize the participation of their child (minor). In the informed consent form filled out by the parents, it was made clear that if they agreed to their child’s participation, they would not be able to know their child’s answers. If the adolescent was of legal age, they signed or put their fingerprint in the informed consent form before agreeing to answer the questions at the health center. The participants decided on the day and time to fill out the questionnaires at a Health Center from the Mexican Government’s Secretariat to protect confidentiality. The questionnaires were self-completed, but if anyone did not feel comfortable reading Spanish, help was provided in filling them out. Refreshments were offered at the time of filling out the questionnaires.

### 2.5. Statistical Analysis

The data were analyzed with the Statistical Package for the Social Sciences (SPSS) version 20 software (SPSS Inc., Armonk, NY, USA), for Windows. Descriptive and inferential statistics were used, the former to figure out the characteristics of the participants by calculating frequencies and percentages as measures of central tendency and dispersion. Multiple linear regression models were conducted using the bootstrap technique to make inference about an estimate for a population parameter on this sample without normal distribution data.

The first model used the enter method and the final model was obtained through backward regression. The independent variables were age, sex, age at menarche/first ejaculation, attitude toward condom use, sexual communication with parents, partner approval of condom use, peer influences, gender role in sexual relations, religious beliefs, ethnic identity, occupation, knowledge about STIs, self-efficacy for condom use, ability to negotiate condom use and intention to use condoms. The dependent variable was condom use.

## 3. Results

The sample consisted of 419 participants aged between 15 and 19 years (M = 17.5, SD = 0.97). The age at menarche/first ejaculation was between 10 and 16 years (M = 12.6, SD = 1.50). The number of years of education was from 5 to 12 (M = 10.00, SD = 1.30), showing the participants were minimally at basic secondary education level (Table 1).

Table 2 presents the socio-demographic information. The proportion of female adolescents (56.8%) was higher than that of male. About half of the participants were in school (50.40%), while the rest were working or had no other occupation, and 63% identified as Catholic.

Table 3 shows the descriptive statistics for the dependent variable. In the participants’ responses for condom use, the lowest mean was 1.25 for the item “I have anal intercourse without using a condom” which corresponded to “never”. In contrast, the highest mean was 3.05 for the item “I insist on condom use when I have sex” and corresponded to “most of the time”. These means stood for the frequency with which they carried a condom with them before possible sexual intercourse or whether they could refuse to have unprotected sex.

The reliability analysis was performed using Cronbach’s alpha to evaluate the extent to which the concept is consistent with what is to be evaluated, so that scores above 0.70 are acceptable. Excellent, good, and acceptable scores were obtained for all scales. None of the variables analyzed had a normal distribution, so regressions were performed using the bootstrap technique (Table 4).

The first regression model with condom use as the dependent variable was significant (F [21,394] = 5.79, *p* < 0.001) and explained 21.6% of the variance. Of the variables included in the first model, attitude toward condom use and the ability to negotiate condom use had a moderate to small effect on condom use. From the above, it can be inferred that the greater the positive attitude (β = 0.16, t = 3.06, *p* = 0.002) toward condom use and the greater the ability to negotiate (β = 0.13, t = 2.49, *p* = 0.013) condom use, the greater the frequency of condom use (Table 5).

The final multiple linear regression model with backward technique is presented. Eleven models were generated from which the following variables were eliminated in the following order: religious beliefs, occupation, partner approval of condom use, age, religious approval of condom use, sex, gender role in sexual relations, intention to use condoms, knowledge about STIs, condom use and pregnancy, and self-efficacy for condom use.

The regression model for the condom use variable was significant (F [4,413] = 24.94, *p* < 0.001) and explained 22.3% of the variance. Age at menarche/first ejaculation had large effect on condom use, while attitude toward condom use, ethnic identity, and the ability to negotiate condom use had a moderate effect. That is, the younger the age at menarche/first ejaculation (β = −1.21, t = −2.08, *p* = 0.038), the greater the value of attitude toward condom use (β = 0.17, t = 3.57, *p* < 0.001), the greater the ethnic identity value (β = 0.21, t = 5.24, *p* < 0.001) and the greater the ability to negotiate condom use value (β = 0.13, t = 3.03, *p* = 0.003), the greater the condom use value (Table 6).

## 4. Discussion

This study determined the relationship between the personal biological (age at menarche/first ejaculation), affective (attitude toward condom use), and cognitive (ability to negotiate condom use) factors along with sociocultural environmental (ethnic identity) factors and the UC of Nahua adolescents. The final model includes variables that positively and negatively affect condom use among Nahua adolescents. Results show that the older the age of first menstruation in the case of females and the older the age of first ejaculation in the case of males, the lower the probability of condom use. The variables of attitude, ethnic identity, and ability to negotiate had positive effects on condom use.

Adolescents who had their menarche/first ejaculation at an older age tended to use condoms less often. This finding supports those of De Almeida and collaborators [25], who found that the age of menarche was a predisposing factor for the initiation of sexual relations, unprotected sex, and pregnancy; adolescents with early menarche tended to initiate their sexual life before the age of 15 years; and that STIs were related to the onset of menarche [26,27]. There may be cultural significance to menarche among the Indigenous population. As girls become women and boys become men, they believe they are ready to have sex [28].

In relation to personal affective factors, it was found that adolescents who assigned greater acceptance to condoms and who believed that they did not interfere with sexual pleasure were more likely to engage in safe sexual behavior and to use a condom consistently and correctly during sexual intercourse. This finding coincides with the study by Guzzo and Hayford [29] who reported that the greater the positive attitude toward condom use, the greater the probability of using a condom for the first vaginal intercourse, both in men and women. Specifically in men, a positive attitude toward condom use increased the probability of use for the first vaginal and anal intercourse.

In terms of ethnic identity, it was found that adolescents who expressed a greater sense of attachment to their ethnic group through exploration and learning activities were more interested in safe sexual behaviors such as the correct and consistent use of condoms during sexual intercourse. This result is congruent with that reported by Meneses-Tirado and collaborators [5], who detailed in a model that adolescents who showed greater commitment to their ethnic group were less likely to have sexual intercourse, and that ethnic identity was a protective factor for safe sex, for a lower number of sexual partners, and for proposing the first sexual relation. However, Nava-Navarro et al. [6] and Bukuluki et al. [7] reported that greater ethnic identity value increased the likelihood of STI presentation. The disparate findings in these studies may be due to inclusion of adults in the study samples.

These data confirm that social and psychological beliefs of ethnicity may be a source of support that promotes healthy development and protects adolescents from exposure to sexual risk; it may also be positively associated with health and risk behaviors with partners [5]. The Nahua community in Mexico is increasing in population and is in permanent contact with society. The community values current access to health services and feels protected by their environment. This belief of social support may influence protective health behaviors, including sexual and reproductive health [30].

Condom use provides numerous potential benefits in terms of sexual health and wellbeing. Condoms function as a physical barrier blocking the transmission of STIs and reducing the spread of HIV/AIDS. Condoms also promote safe sex practices, thereby leading to healthier relationships and reducing unwanted pregnancy. It was found that the greater the perceived ability of the adolescent to be able to discuss or propose condom use in sexual practice, the greater the frequency of safe sexual behavior. This result is similar to that reported by Rosenstock et al. [31], who found that among adolescents of ethnic origin, confidence in negotiating condom use with partners was an important predictor of actual condom use. This implied that girls who lacked confidence in negotiating condom use with their sexual partners and were exposed to unprotected sex had an increased risk of contracting STIs, including HIV/AIDS, and vulnerability to adverse sexual and reproductive health outcomes such as unwanted pregnancies. It is well known that indigenous adolescents are at higher risk for unplanned pregnancy, as it is higher than among adolescents who do not speak an Indigenous language [32]. Santos and collaborators [33] found a similar result in which communication with the partner and the ability to negotiate were positive aspects that encouraged consistent condom use.

The above may be due to a person’s ability to anticipate the consequences of their actions and to establish goals that guide them, since a desirable event motivates a person to carry out what is of interest to them. This ability is related to the socialization process of members of a group. Through this socialization process, interpersonal demands and sanctions are internalized and direct behavior, whether as a couple or in society, which finally allows them to learn about themselves and the world, evaluate themselves, and, if necessary, change their behavior [13].

### 4.1. Limitations

Methodologically, the cross-sectional design did not allow us to show causality. Likewise, the statistical analysis used was limited to the relationships proposed by the goal; considering the number of variables, other statistical techniques could provide more information on the associations of the variables; for example, structural equation modeling. Finally, the information was obtained in only one community in the state of Puebla, so the results cannot be generalized to the sixty-two ethnic groups that exist in the country until they are assessed in other communities. 

Furthermore, this study has some limitations regarding the influence of desirability bias on perceived condom use. Adolescents may overestimate their sexual behavior because it is considered undesirable or stigmatized in Indigenous communities. This predisposition has significant implications for the accuracy of self-reported behavior. Further limitations include limited access to condoms, which was not measured in this study. When examining condom use, access to condoms is critical because cost may be a significant barrier to condom access, particularly for participants in this study lacking sufficient financial resources.

In addition, we did not examine individual attitudes toward condom use in terms of anticipation of sexual behavior. It is possible that adolescents are more likely to consider condoms important when they know that sexual acts may occur. Another limitation may be that we considered condom use only as a behavior for STI/AIDS prevention. It is important to recognize that other forms of contraception may be in use, as some women who prioritize pregnancy prevention may disregard STI risk and therefore may use condoms less frequently. Finally, we did not consider the potential influence of prior STI infection or prior HIV testing on condom use behavior, and this may alter perceptions and understanding of the importance of condom use as STI protection.

### 4.2. Recommendations

A continued focus on Nahua adolescents is imperative due to the scarcity of scientific research among this disadvantaged group. Theory-based behavioral and psychoeducational interventions that improve the sexual and reproductive health of Nahua adolescents require incorporation of findings from this study. Other considerations include a qualitative or mixed approach enriched by different areas such as sociology, anthropology, and psychology, to obtain more complete data on safe sexual behavior, condom use, and safe sex. Expansion of the statistical analysis, such as path analysis from the variables that were predictive in the model, could help to explain the interaction of the proposed concepts and serve as a reference for future research.

## 5. Conclusions

The results of this research serve as a basis for generating interventions considering the cultural values of Indigenous Nahua adolescents to promote sexual health. It is important to highlight that these results will be valuable to provide culturally sensitive counseling within programs aimed at prevention of early pregnancies, with an emphasis on vulnerable populations of Indigenous adolescents.

## Figures and Tables

**Table 1 children-10-00921-t001:** General identification data.

Variable	M	Mdn	SD	Min	Max
Age	17.50	18	0.97	15	19
Age at menarche/first ejaculation	12.60	12	1.50	10	16
Schooling years	10.00	10	1.30	5	12

*n* = 419; M = mean; Mdn = median, SD = standard deviation; Min = minimum; Max = maximum.

**Table 2 children-10-00921-t002:** Socio-demographic frequencies.

Variable	Frequencies	%
Sex		
Male	181	43.2
Female	238	56.8
Occupation		
Student	211	50.4
Worker	111	26.5
Student and Worker	87	20.8
None	10	2.4
Religion		
Catholic	264	63.0
Christian	25	6.0
Protestant	2	0.5
Jehovah’s Witness	57	13.6
None	34	8.1
Other	37	8.8

*n* = 419; *f* = frequencies; % = percentage.

**Table 3 children-10-00921-t003:** Condom use instrument questions.

Items	Mean	Mdn	SD
I insist on condom use when I have sex.	3.05	3.00	1.00
I stop foreplay (such as touching/kissing/gagging) in time to put on a condom or to have my partner put on a condom.	2.74	3.00	1.07
If I know that in an encounter with someone, we may have sex, I carry a condom with me.	2.72	3.00	1.10
If I get carried away by the passion of the moment, I have sex without using a condom.	1.52	1.00	0.78
I have oral sex without using protective barriers, such as a condom or latex cover.	1.43	1.00	0.82
If my partner insists on having sex without using a condom, I refuse to have sex.	2.95	3.00	1.09
I have anal intercourse without using a condom.	1.25	1.00	0.62

*n* = 419, Mdn = median, SD = standard deviation, 1 = never, 2 = sometimes, 3 = most of the time, 4 = always.

**Table 4 children-10-00921-t004:** Reliability of the instruments and the normality test.

Instrument	Items	α	D*^a^*	*p*-Value
Attitude toward condom use	7	0.76	0.07	0.001
Knowledge about STIs	24	0.81	0.14	0.001
Self-efficacy for condom use	12	0.86	0.05	0.004
Assertiveness in condom use	7	0.82	0.08	0.001
Intention to use condoms	4	0.84	0.13	0.001
Sexual communication with parents	9	0.92	0.07	0.001
Partner approval of condom use	4	0.68	0.10	0.001
Peer influences	5	0.85	0.17	0.001
Gender role in sexual relations	24	0.88	0.18	0.001
Religious beliefs	19	0.90	0.09	0.001
Ethnic identity	17	0.92	0.08	0.001
Condom use	7	0.68	0.11	0.001

*n* = 419, α = Cronbach’s alpha.

**Table 5 children-10-00921-t005:** Personal, interpersonal, and sociocultural factors on condom use (initial model).

Variables	β	SD	*t*	*p*-Value
Constant	61.47	20.25	3.03	0.003
Age	−0.19	−0.90	−0.21	0.833
Male	−1.27	2.27	−0.56	0.576
Age at menarche/first ejaculation	−0.88	0.71	−1.23	0.220
Attitude toward condom use	0.16	0.05	3.06	0.002
Sexual communication with parents	0.06	0.03	1.84	0.066
Partner approval of condom use	0.01	0.04	0.30	0.762
Peer influences	−0.04	0.04	−0.92	0.360
Gender role in sexual relations	−0.04	0.07	−0.56	0.578
Religious beliefs	−0.02	0.05	−0.30	0.762
Religious approval (Totally agree)				
Totally disagree	−0.09	3.80	0.02	0.981
Disagree	−5.22	3.73	−1.40	0.162
Neutral	−1.25	2.75	−0.45	0.650
Agree	−2.08	2.88	−0.72	0.470
Ethnic identity	0.12	0.06	1.90	0.058
Occupation (None)				
Student	−0.99	5.53	−0.18	0.858
Employed	−1.60	5.64	−0.28	0.777
Both	0.95	5.64	0.17	0.867
Knowledge about STIs	0.06	0.05	1.13	0.257
Self-efficacy for condom use	−0.08	0.05	−1.54	0.123
Ability to negotiate condom use	0.13	0.05	2.49	0.013
Intention to use condoms	0.03	0.03	0.80	0.422

Dependent variable = condom use, β = beta, SD = standard deviation.

**Table 6 children-10-00921-t006:** Predictors of Condom Use (Final model).

Variables	β	SD	*t*	*p*-Value
Constant	58.02	8.68	6.68	0.001
Age at menarche/first ejaculation	−1.21	0.58	−2.08	0.038
Attitude toward condom use	0.17	0.05	3.57	0.001
Ethnic identity	0.21	0.07	5.24	0.001
Ability to negotiate condom use	0.13	0.04	3.03	0.003

Dependent variable = condom use, β = beta, SD = standard deviation.

## Data Availability

Not applicable.

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
