# Peer review of "Personal, Interpersonal, and Sociocultural Factors of Condom Use in Rural Indigenous Nahuas Adolescents in Mexico"

_children, 2023, doi:10.3390/children10060921_

Round 1
Reviewer 1 Report (New Reviewer)
Dear authors,
I was glad having the opportunity to revise the above-mentioned manuscript entitled “Personal, Interpersonal, and Sociocultural Factors of Condom Use in Rural Indigenous Nahuas Adolescents in Mexico” by Raquel Alicia Benavides-Torreset al.
Despite the limitations and possible mis-perfections of the study, my impression is that the particular well written and presented study should be considered for publication in your prestigious journal.
Additionally, similar publications by other scientific groups in the particular field should also be encouraged, as protective role of condom use in sexually transmitted diseases is of great importance.
I do have some minor comments for the authors:
Lines 42-48
Please re-write the paragraph, giving more detailed information. The sentence “It has been reported that the female sex does not have the ability to use a condom or to put it on her partner” does not make sense. You can also add some sentences regarding other common sexually transmitted infections that individuals could potentially be protected by the condoms.
Discussion session
Lines 267-274
Please re-write the sentence “This may be due to the sociocultural significance of menarche among the indigenous population, since menarche means that girls become women and boys become men, giving them the freedom to have sexual relations”.
Lines 307-312
Please add a paragraph presenting additional potential benefits from condom use in general; explaining the effectiveness of the consistent use and the positive effect in acquiring sexually transmitted infections and the positive long-term effect in already positive women in terms of clearance of the disease.
Minor English language editing is needed.
Author Response
Dear reviewer, Thank you very much for your feedback.
The following is the response to your comments
I do have some minor comments for the authors:
Lines 42-48
Please re-write the paragraph, giving more detailed information. The sentence “It has been reported that the female sex does not have the ability to use a condom or to put it on her partner” does not make sense.
Response: Re-write as “Women reportedly have more difficulty reaching agreement with their partners about condom use than men. They also have difficulty with correct condom use such as placement of condoms on their partners”
You can also add some sentences regarding other common sexually transmitted infections that individuals could potentially be protected by the condoms.
Response: Add “Condoms are proven to be an effective barrier method for STI prevention. Condoms protect again STIs including chlamydia, gonorrhea, syphilis, and HIV/AIDS”
Discussion session
Lines 267-274
Please re-write the sentence “This may be due to the sociocultural significance of menarche among the indigenous population, since menarche means that girls become women and boys become men, giving them the freedom to have sexual relations”.
Response: Re-write “There may be cultural significance to menarche among the indigenous population. As girls become women and boys become men, they believe they are ready to have sex”
Lines 307-312
Please add a paragraph presenting additional potential benefits from condom use in general; explaining the effectiveness of the consistent use and the positive effect in acquiring sexually transmitted infections and the positive long-term effect in already positive women in terms of clearance of the disease.
Response: Add “Condom use provides numerous potential benefits in terms of sexual health and wellbeing. Condoms act as a physical barrier blocking the transmission of STIs and reducing the spread of HIV/AIDS. Condoms also promote safe sex practices there byleading to healthier relationships and reducing unwanted pregnancy”.
Reviewer 2 Report (New Reviewer)
The manuscript is clearly written, well organized, sufficiently detailed, and supported by relevant literature. The theories are well presented in the introduction/background, but could be reconsidered in the discussion/conclusion. There are typos throughout that should be addressed.
Limitations should be expanded. First, these are all self-reported behaviors. We know that people over-estimate behaviors that are determined desirable. If the study was presented as an HIV/STI prevention study then likely condom use was overestimated. Second, while presented in the background, "environment" was not fully examined in the methods. Were study participants able to access condoms? In the US condoms are very expensive. Unless you are at a clinic where they are provided for free, they are costly to acquire. Furthermore, what about locations of sexual engagements, does this change condom use? If young people are anticipating a sexual act with planning are they more likely to be prepared to use condoms (both having condoms on hand and being ready to negotiate their use)? Third, regarding females, did you consider alternative contraception use? While this study focused on HIV/STI prevention, this may be a secondary concern for young women who are focused on pregnancy prevention. Did the use of other contraception alter condom use? Finally, what about prior experiences with STIs/HIV? Did prior STI infection or prior HIV testing (suspected infection) alter condom use patterns?
These weren't measured as part of the study but should be discussed as these are drivers of condom use. HIV/STI prevention cannot be taken out of context. The full scope of health considerations and structural barriers should be considered.
Author Response
Dear reviewer, Thank you very much for your feed back.
The following is the response to your comments
The manuscript is clearly written, well organized, sufficiently detailed, and supported by relevant literature. The theories are well presented in the introduction/background, but could be reconsidered in the discussion/conclusion. There are typos throughout that should be addressed.
Response: Typos were corrected
Limitations should be expanded. First, these are all self-reported behaviors. We know that people over-estimate behaviors that are determined desirable. If the study was presented as an HIV/STI prevention study then likely condom use was overestimated. Second, while presented in the background, "environment" was not fully examined in the methods. Were study participants able to access condoms? In the US condoms are very expensive. Unless you are at a clinic where they are provided for free, they are costly to acquire. Furthermore, what about locations of sexual engagements, does this change condom use? If young people are anticipating a sexual act with planning are they more likely to be prepared to use condoms (both having condoms on hand and being ready to negotiate their use)? Third, regarding females, did you consider alternative contraception use? While this study focused on HIV/STI prevention, this may be a secondary concern for young women who are focused on pregnancy prevention. Did the use of other contraception alter condom use? Finally, what about prior experiences with STIs/HIV? Did prior STI infection or prior HIV testing (suspected infection) alter condom use patterns?
Response: Limitations were explained “Furthermore, this study has some limitations regarding the influence of desirability bias on perceived condom use. Adolescents may overestimate their sexual behavior because it is considered undesirable or stigmatized in indigenous communities. This predisposition has significant implications concerning the accuracy of self-reported behavior. Further limitations include limited access to condoms, which was not measured in this study. When examining condom use, access to condoms is critical because cost may be a significant barrier to condom access, particularly for participants in this study lacking sufficient financial resources.
Addition, we did not examine individual attitudes toward condom use in terms of anticipation of sexual behavior. It is possible that adolescents are more likely to consider condoms important when they know that sexual acts may occur. Another limitation may be that we considered condom use only as a behavior for STI/AIDS prevention. It is important to recognize that other forms of contraception may be in use, as some women who prioritize pregnancy prevention may disregard STI risk and therefore may use fewer condoms. Finally, we did not consider the potential influence of prior STI infection or prior HIV testing on condom use behavior and this may alter perceptions and understanding of the importance of condom use as STI protection”
These weren't measured as part of the study but should be discussed as these are drivers of condom use. HIV/STI prevention cannot be taken out of context. The full scope of health considerations and structural barriers should be considered.
This manuscript is a resubmission of an earlier submission. The following is a list of the peer review reports and author responses from that submission.
Round 1
Reviewer 1 Report
This is a strong study on an important topic. Overall its well written. I recommend avoiding very long sentences. They are hard for the reader to understand. I have listed several instances below of long sentences that should be broken into 2 or 3 sentences each.
Long sentences: 45-49; 73-76; 103-108; 155-159; 240-246
Line 35 - Is there only 1 indigenous nation in Mexico? Are all indigenous people Nahua? It would help the reader to understand the context if you can explain this. In the US there are many indigenous nations so we wouldn't refer to indigenous people by the name of one nation/band/tribe.
The word present in line 245 doesn't make sense so seems like a mistake
Some points to clarify: 120-121, do you mean that participants who spoke an IL or if their parents spoke an IL was a way to determine that they were members of an IL?
Did participants have to be fluent in Spanish too? or English?
153-159- Clarify the recruitment and consent process. Then clarify where and how data were collected. How did you get from the 35 identified people (line 118) to the 419 actual participants? When you identified the random people, did you contact them to set up the home visit? Were the home visits cold calls? Were home visits done to obtain participant and parent assent and consent? By whom? When were participants screened for inclusion criteria? At a home visit? with parents present?
Was data collected in the home too? Or did participants come to a health center? What language was data collected in? How were the instruments translated (if they were)?
Line 185 - there are 12 instruments in the table. Five instruments are described in section 2.2. Section 2.2 should include description of the 12 instruments in the table.
Discussion
Delete 226 -228 this sentence is not supported by your data. The phrase "gives them the freedom" is judgmental. Perhaps there is a cultural norm that people initiate a sexual life after biological maturity, but as its phrased, it sounds derogatory to young people. Also the word irresponsible is judgmental. There are many reasons why people don't use condoms as is described in the introduction. Judging people negatively for not using condoms is not supported by your data but also not helpful in the efforts to encourage more condom use.
Lines 237-250 - was there anything specific to Nahua culture that might explain how ethnic identity was protective? In line 41, you wrote that sociocultural environmental factors influence sexual behaviors. Can you describe this influence in this paragraph.
Line 249-250 - this is a new idea that hasn't been part of this paper until this point. Why is this sentence here, what does it mean for your study and was it related to your results?
Line 270 Is another limitation the inclusion of only one indigenous nation?
Author Response
Response: Many thanks for the valuable and timely comments on the manuscript.
This is a strong study on an important topic. Overall its well written. I recommend avoiding very long sentences. They are hard for the reader to understand. I have listed several instances below of long sentences that should be broken into 2 or 3 sentences each.
Long sentences: 45-49; 73-76; 103-108; 155-159; 240-246
Response: The long sentences were reduced
Line 35 - Is there only 1 indigenous nation in Mexico? Are all indigenous people Nahua? It would help the reader to understand the context if you can explain this. In the US there are many indigenous nations so we wouldn't refer to indigenous people by the name of one nation/band/tribe.
Response: The wording was changed to explain that the Nahua are one of the largest communities in Mexico. This implies that they are not the only ones.
The word present in line 245 doesn't make sense so seems like a mistake
Response: Error corrected
Some points to clarify: 120-121, do you mean that participants who spoke an IL or if their parents spoke an IL was a way to determine that they were members of an IL?
Did participants have to be fluent in Spanish too? or English?
Response: The wording has been changed to make it clearer and we add Spanish proficiency was not a criterion for inclusion.
153-159- Clarify the recruitment and consent process.
Response: Clarified
Then clarify where and how data were collected. How did you get from the 35 identified people (line 118) to the 419 actual participants?
Response: We add the following to clarify: “Single stage cluster sampling was used considering the total number of blocks in the neighborhood and the total number of adolescents per block, so 32 block were randomly selected to achieve the sample size”
When you identified the random people, did you contact them to set up the home visit? Were the home visits cold calls? Were home visits done to obtain participant and parent assent and consent? By whom? When were participants screened for inclusion criteria? At a home visit? with parents present? Was data collected in the home too? Or did participants come to a health center? What language was data collected in? How were the instruments translated (if they were)?
Response: All this information was added in the data collection section.
Line 185 - there are 12 instruments in the table. Five instruments are described in section 2.2. Section 2.2 should include description of the 12 instruments in the table.
Response: Added the wording of all missing instruments in the measurement section.
Discussion
Delete 226 -228 this sentence is not supported by your data. The phrase "gives them the freedom" is judgmental. Perhaps there is a cultural norm that people initiate a sexual life after biological maturity, but as its phrased, it sounds derogatory to young people. Also the word irresponsible is judgmental. There are many reasons why people don't use condoms as is described in the introduction. Judging people negatively for not using condoms is not supported by your data but also not helpful in the efforts to encourage more condom use.
Response: You are correct, so these lines have been deleted.
Lines 237-250 - was there anything specific to Nahua culture that might explain how ethnic identity was protective? In line 41, you wrote that sociocultural environmental factors influence sexual behaviors. Can you describe this influence in this paragraph.
Response: This was added: “Currently, the Nahua community in Mexico has been growing and is in permanent contact with society. They value current access to health services and feel protected by their community, this perception of social support may influence protective health behaviors, including sexual and reproductive health.”
Line 249-250 - this is a new idea that hasn't been part of this paper until this point. Why is this sentence here, what does it mean for your study and was it related to your results?
Response: These lines have been deleted.
Line 270 Is another limitation the inclusion of only one indigenous nation?
Response: Yes, this is a limitation since it cannot be generalized to the other 62 indigenous groups that exist in Mexico. We add to clarify this.

Reviewer 2 Report
Thank you for the opportunity to review your manuscript. I found it to be well-written overall and engaging. Section 4 in particular is written beautifully and clearly, with helpful references to other literature on the subject. There are several instances where typographical errors were made, or perhaps mistranslation. I recognize that I may also be trying to apply my own colloquial form of speech. I look forward to reading more work by your group.
Line 13: depluralize "objectives" and change the second "objective" to a synonym like "goal"
Line 19: please give the definition you used to call someone a "student" (see line 169)
Line 20: it's very interesting to me you used menarche and first ejaculation in parallel; were the findings any different if you separated them by sex?
Line 28: the first line of the introduction is a bit reductive; consider rephrasing it (eg, "Even decades into the epidemic, HIV infection is still a huge public health issue.")
Line 45: change "biological sex at birth" to "sex assigned at birth;" change "a woman or a man" to "a girl or a boy," remembering that sometimes babies can be born intersex
Line 47: are female condoms not available?
Line 58: the parenthetical "as if that were not enough" is unnecessary
Line 73: the clarifying comment "(cognitive)" should go after the word "mental"
Lines 95-97: "biological," "affective," and "cognitive" are adjectives, so please add a noun at the end of this sentence (eg, elements)
Line 103: rephrase the beginning of the sentence to "... importance for nurses to study attitudes..."
Line 120: I don't understand whether literacy was an inclusion criterion (since illiteracy is mentioned right after)
Line 149: I don't believe you've used first person plural prior to this, so consider rephrasing this sentence in the passive voice (ie, "... the official letter issued by the FAEN was delivered to the Centro de Salud...")
Line 157: if illiteracy was an inclusion criterion (see line 120), then how were informed consent forms signed?
Lines 165 and 184: my apologies, I do not know what the bootstrap technique is - could you please define it for non-statisticians?
Line 182: similarly, is it possible to explain Cronbach's alpha without distracting from the paper?
Line 196: please move "final" to before "multiple"
Line 265: do you mean "coupling" rather than "couple"?
Line 277: the final period in the sentence is missing
Lines 288-289: can you offer an example of a possible such intervention based on this project?
Lines 294-304: unless advised otherwise by the editorial staff, I would use contributors' initials when listing who did what - as it stands, it is far too easy to get "lost" in all of the text
Line 308: there is a missing end parenthesis
Lines 314-363: this comment is only regarding internet documents/pages (not articles) - ideally, you would list the date the websites were accessed, since sometimes sites will update the file without changing the URL or, alternatively, the URL becomes nonfunctional
Author Response
Thank you for the opportunity to review your manuscript. I found it to be well-written overall and engaging. Section 4 in particular is written beautifully and clearly, with helpful references to other literature on the subject. There are several instances where typographical errors were made, or perhaps mistranslation. I recognize that I may also be trying to apply my own colloquial form of speech. I look forward to reading more work by your group.
Response: Thank you very much, it is very kind of you and very helpful.
Line 13: depluralize "objectives" and change the second "objective" to a synonym like "goal"
Response: Changed
Line 19: please give the definition you used to call someone a "student" (see line 169)
Response: Changed
Line 20: it's very interesting to me you used menarche and first ejaculation in parallel; were the findings any different if you separated them by sex?
Response: We did not analyze that but may be interesting to see in a future paper.
Line 28: the first line of the introduction is a bit reductive; consider rephrasing it (eg, "Even decades into the epidemic, HIV infection is still a huge public health issue.")
Response: rephrased
Line 45: change "biological sex at birth" to "sex assigned at birth;" change "a woman or a man" to "a girl or a boy," remembering that sometimes babies can be born intersex
Response: Changed
Line 47: are female condoms not available?
Response: Female condoms are not really available in Mexico
Line 58: the parenthetical "as if that were not enough" is unnecessary
Response: Deleted
Line 73: the clarifying comment "(cognitive)" should go after the word "mental"
Response: Deleted cognitive
Lines 95-97: "biological," "affective," and "cognitive" are adjectives, so please add a noun at the end of this sentence (eg, elements)
Response: The noun factors was added at the end of the sentence.
Line 103: rephrase the beginning of the sentence to "... importance for nurses to study attitudes..."
Response: Changed
Line 120: I don't understand whether literacy was an inclusion criterion (since illiteracy is mentioned right after)
Response: We corrected because it was not. “Spanish proficiency was not a criterion for inclusion”.
Line 149: I don't believe you've used first person plural prior to this, so consider rephrasing this sentence in the passive voice (ie, "... the official letter issued by the FAEN was delivered to the Centro de Salud...")
Response: Changed
Line 157: if illiteracy was an inclusion criterion (see line 120), then how were informed consent forms signed?
Response: Changed to “they signed or put their fingerprint”
Lines 165 and 184: my apologies, I do not know what the bootstrap technique is - could you please define it for non-statisticians?
Response: We clarify “Multiple linear regression models were conducted using the bootstrap technique to make inference about an estimate for a population parameter on this sample with not normal distribution data”
Line 182: similarly, is it possible to explain Cronbach's alpha without distracting from the paper?
Response: We modify “The reliability analysis was performed using Cronbach's alpha to evaluate the extent to which the concept is consistent with what is to be evaluated, so that scores above .70 are acceptable”
Line 196: please move "final" to before "multiple"
Response: Changed
Line 265: do you mean "coupling" rather than "couple"?
Response: Changed to “This capacity is related to the socialization process as members of a group”
Line 277: the final period in the sentence is missing
Response: added
Lines 288-289: can you offer an example of a possible such intervention based on this project?
Response: added
Lines 294-304: unless advised otherwise by the editorial staff, I would use contributors' initials when listing who did what - as it stands, it is far too easy to get "lost" in all of the text
Response: Changed
Line 308: there is a missing end parenthesis
Response: added
Lines 314-363: this comment is only regarding internet documents/pages (not articles) - ideally, you would list the date the websites were accessed, since sometimes sites will update the file without changing the URL or, alternatively, the URL becomes nonfunctional
Response: added

Reviewer 3 Report
Summary
This original study conducted a cross-sectional study to investigate the predictor of condom use among rural indigenous adolescents in Mexico. This study is interesting and can contribute to the current sexual behaviour interventions, but I have a few comments below.
Major Issues
Introduction
1. Can the authors provide a reference for the statement “This incidence increases in communities of indigenous origin.”?
2. I found “as they may explore their sexuality without considering the risk and consequences of sexually transmitted infections (STIs) or unwanted pregnancy.” stigmatised, I do not think sexuality has anything to do with STIs and unwanted pregnancy. The authors need to use non-stigmatising language.
3. Also, there are differences between sexual behaviours and condom use. The authors jumped to “Among the factors related to condom use” right after the paragraph on sexual behaviours matters. Personally, I consider this to be disconnected.
4. Generally, I think the text flow of the first part of the introduction must be further improved.
5. There are differences between attitudes and beliefs. The authors mixed the conceptions of negative attitudes and negative beliefs about the use of using condoms.
Methods
1. How did the author proceed with the multiple linear regression? Forward or backward method? Such as sex, gender, sexualities, etc,. The authors should add more information on model selection and their criteria in section 2.5
Results
1. Can the authors also provide a table of study population characteristics to describe the study population better for improved transparency?
2. For table 2, can the authors also include the 95%CI for alpha estimation? Also, can the authors also elaborate on the results of the relatively low reliability of the instrument “Partner approval of condom use” and “Condom use”?
3. Can the authors also provide 95%CI for the estimated coefficient in Table 3 and Table 4?
4. The authors explained their statistical methods in the results section, such as the model selection procedure. However, such information should belong to the Methods section.
5. Can the authors also provide information on the goodness of fit of the models in the results, such as AIC?
Discussion
1. The authors should point out the direction of the effects from the predictors in the general discussion. For example, age was the only predictor that was estimated to have a negative effect on condom use.
Minor Issues
1. I would suggest the authors update the title based on the PICOT approach to make it clearer and more straightforward. Also, if possible, add the study time in the title. Also, given that the authors only collect data from one community, I would suggest enriching the title's details to better inform the readers.
2. can the authors also provide the reference number for ethical approval?
Author Response
Summary
This original study conducted a cross-sectional study to investigate the predictor of condom use among rural indigenous adolescents in Mexico. This study is interesting and can contribute to the current sexual behaviour interventions, but I have a few comments below.
Response: Thank you very much for all your suggestions as they were useful to improve the manuscript.
Major Issues
Introduction
- Can the authors provide a reference for the statement “This incidence increases in communities of indigenous origin.”?
Response: Reference was added
- I found “as they may explore their sexuality without considering the risk and consequences of sexually transmitted infections (STIs) or unwanted pregnancy.” stigmatised, I do not think sexuality has anything to do with STIs and unwanted pregnancy. The authors need to use non-stigmatising language.
Response: Sentence was modified, “These factors increase the vulnerability of adolescents, because at this time they often take risks in their sexual relations regardless of the consequences, which include sexually transmitted infections (STIs) and unwanted pregnancies”
- Also, there are differences between sexual behaviours and condom use. The authors jumped to “Among the factors related to condom use” right after the paragraph on sexual behaviours matters. Personally, I consider this to be disconnected.
Response: We added before “One of the most frequent risks that adolescents present, is having sex without using a condom”.
- Generally, I think the text flow of the first part of the introduction must be further improved.
Response: Improved the presentation of the introduction
- There are differences between attitudes and beliefs. The authors mixed the conceptions of negative attitudes and negative beliefs about the use of using condoms.
Response: This was corrected
Methods
- How did the author proceed with the multiple linear regression? Forward or backward method? Such as sex, gender, sexualities, etc,. The authors should add more information on model selection and their criteria in section 2.5
Response: We added “Multiple linear regression models were conducted using the bootstrap technique to make inference about an estimate for a population parameter on this sample with not normal distribution data. The initial models used the enter method and the final model was obtained through backward regression”.
Results
- Can the authors also provide a table of study population characteristics to describe the study population better for improved transparency?
Response: We added two tables one with continues data and the other of the frequencies of categorical data.
- For table 2, can the authors also include the 95%CI for alpha estimation? Also, can the authors also elaborate on the results of the relatively low reliability of the instrument “Partner approval of condom use” and “Condom use”?
Response: Sorry but when the analysis was performed, we did not request the CI therefore we can not add that information. At this moment we do not have the data base available for further analysis.
- Can the authors also provide 95%CI for the estimated coefficient in Table 3 and Table?
Response: Sorry but when the analysis was performed, we did not request the CI therefore we cannot add that information. At this moment we do not have the data base available for further analysis.
- The authors explained their statistical methods in the results section, such as the model selection procedure. However, such information should belong to the Methods section.
Response: We included more information on the Statistical Analysis section.
- Can the authors also provide information on the goodness of fit of the models in the results, such as AIC?
Response: Sorry but we did not calculate the goodness of fit, therefore we can not include that information. At this moment we do not have the data base available for further analysis.
Discussion
- The authors should point out the direction of the effects from the predictors in the general discussion. For example, age was the only predictor that was estimated to have a negative effect on condom use.
Response: We pointed out the direction of the effect on the first paragraph of the discussion.
Minor Issues
- I would suggest the authors update the title based on the PICOT approach to make it clearer and more straightforward. Also, if possible, add the study time in the title. Also, given that the authors only collect data from one community, I would suggest enriching the title's details to better inform the readers.
Response: Title was changed
- can the authors also provide the reference number for ethical approval?
Response: Ethical approval number of the research is presented on the Institutional Review Board statement section.

Reviewer 4 Report
This is an interesting and well-written paper. The findings are not surprising as they seem to confirm findings of at least one other study. What is puzzling is that it contradicts findings related to STIs. Can the authors explain this finding in the discussion? Can the authors further explain "However, for some youth, the interpretation of a strong ethnic identity may not be positive.", as this seems to contradict their findings. It is not clear to me from the discussion how important the authors think that the strength of indigenous identification is compared to other factors in the use of condoms, although it does make suggestion which seem appropriate for this population. Finally, do they have any similar data from non-indigenous people in the same age range? If yes, they might point out the similarities or differences in the discussion.
Author Response
This is an interesting and well-written paper.
Thank you very much for your observations there were valuable.
The findings are not surprising as they seem to confirm findings of at least one other study. What is puzzling is that it contradicts findings related to STIs. Can the authors explain this finding in the discussion?
Response: We explained this on the discussion “These differences may be since the population of these studies was different as they also included adults in the reports”.
Can the authors further explain "However, for some youth, the interpretation of a strong ethnic identity may not be positive.”, as this seems to contradict their findings. It is not clear to me from the discussion how important the authors think that the strength of indigenous identification is compared to other factors in the use of condoms, although it does make suggestion which seem appropriate for this population.
Response: We did correct that. “These data confirm that social and psychological perceptions of ethnicity may be a source of support that promotes healthy development and protects adolescents from exposure to sexual risk and may also be positively associated with health and risk behaviors with partners [4]. Currently, the Nahua community in Mexico has been growing and is in permanent contact with society. They value current access to health services and feel protected by their community, this perception of social support may influence protective health behaviors, including sexual and reproductive Health.”
Finally, do they have any similar data from non-indigenous people in the same age range? If yes, they might point out the similarities or differences in the discussion.
Response: We improved the discussion and recommendations sections including some differences with non-indigenous population. “It is well known that in the case of unplanned pregnancies they are at a greater disadvantage since the figure is higher than that of young people who do not speak an indigenous language” and “Finally, it is important to highlight that these results will be valuable to provide culturally sensitive counseling to develop programs aimed at the prevention and eradication of early pregnancies, with emphasis on indigenous girls and adolescents because they are considered to be more vulnerable.”

Round 2
Reviewer 1 Report
Thank you for your attention to my comments in your revision. I appreciate the detail you added to recruitment and where data collection was conducted.
It is still unclear if data were collected at home in three home visits or at the health center. Maybe you collected data at both places?
Also, can you include how many the percent of teens who agreed to participate - how many home visits were conducted and how many families agreed to participate? Did you offer a stipend to participants?
I am impressed that you were able to recruit an adequate sample. Going door to door to ask indigenous families to allow their teens to participate in a study to increase condom use just sounds extraordinary. I would think that families would mostly not be interested. Also, how could you protect the teens' privacy regarding their sexual activity status? Even if you asked that in private, their participation in the study would convey to parents that they were sexually active.
Author Response
|
All manuscript sections were improved including English language and style quality. |
( ) |
( ) |
REVIEWER: It is still unclear if data were collected at home in three home visits or at the health center. Maybe you collected data at both places?
RESPONSE: Participants were recruited at their homes, where they were visited up to three times to invite them to participate. Once located and if they met the eligibility criteria, they were scheduled an appointment at the health center where the questionnaires were filled out. This was done to protect the confidentiality of the participants' information regarding sexual issues.
We modify as following:
“Data collection was conducted over a maximum of three visits to their homes, with an explanation to the parent/guardian as well as to the adolescent(s) of the importance and objectives of the study… The participant decided on the day and time to fill out the questionnaires at the Health Center to protect the confidentiality.” (lines 168-181)
REVIEWER: Also, can you include how many the percent of teens who agreed to participate –
RESPONSE: We do not have exact information on how many people accepted to participate, but a non-response rate of 20% was considered, which was not exceeded.
We modify as following:
“The sample size, calculated using the statistical package nQuery Advisor, with a significance level of .05, a correlation coefficient of .10, and a power of 80%, was adequate with an estimated of a non-response rate of 20%.” (line 110)
REVIEWER: how many home visits were conducted and how many families agreed to participate?
RESPONSE: A maximum of three visits were made to invite them to participate. (line 169)
REVIEWER: Did you offer a stipend to participants?
RESPONSE: Refreshments were offered at the time of filling out the questionnaires. (line 180)
REVIEWER: I am impressed that you were able to recruit an adequate sample. Going door to door to ask indigenous families to allow their teens to participate in a study to increase condom use just sounds extraordinary. I would think that families would mostly not be interested.
RESPONSE: It should be noted that the people of this community receive support from the Mexican Government's Health Secretariat, so they are very interested in participating in activities organized by the health care centers to which they belong.(line 178)
REVIEWER: Also, how could you protect the teens' privacy regarding their sexual activity status? Even if you asked that in private, their participation in the study would convey to parents that they were sexually active.
RESPONSE: In the informed consent filled out by the parents, it was made clear that if they agreed to their child's participation, they would not be able to know their child's answers. (line 173)
Reviewer 3 Report
Dear Authors,
Thank you for your efforts on the revision, congratulations from me. However, I have to flag my concerns about the research ethics of your works on data storage for scientific research, as it is commonly required to store your raw data for 10 years after the publication to ensure data and study quality. I have checked the journal's requirements for data storage:
"
- Raw data should preferably be publicly deposited by the authors before submission of their manuscript. Authors need to at least have the raw data readily available for presentation to the referees and the editors of the journal, if requested. Authors need to ensure appropriate measures are taken so that raw data is retained in full for a reasonable time after publication.
"
According to your rebuttal letter, you have claimed that "At this moment we do not have the data base available for further analysis". I have flagged my ethical concern to the editors regarding data quality and data storage. Please follow the international consensus on research data storage. If you are able to access the data again, and ensure its storage, i am very happy to review for you again.
Goodluck.
Author Response
All manuscript sections were improved including English language and style quality.
REVIEWER: I have to flag my concerns about the research ethics of your works on data storage for scientific research, as it is commonly required to store your raw data for 10 years after the publication to ensure data and study quality. I have checked the journal's requirements for data storage:
"Raw data should preferably be publicly deposited by the authors before submission of their manuscript. Authors need to at least have the raw data readily available for presentation to the referees and the editors of the journal, if requested. Authors need to ensure appropriate measures are taken so that raw data is retained in full for a reasonable time after publication."
According to your rebuttal letter, you have claimed that "At this moment we do not have the data base available for further analysis". I have flagged my ethical concern to the editors regarding data quality and data storage. Please follow the international consensus on research data storage. If you are able to access the data again, and ensure its storage, i am very happy to review for you again.
RESPONSE: I apologize that my answer was not very clear and was misinterpreted. What I meant to say is that I do not have permission from the IRB committee (IRB registered on the Office of Human Research Protection) to perform additional analysis to that requested in the initially approved protocol. Perhaps this analysis can be included in a future article for which permission will be requested and a secondary data analysis can be performed for this purpose.
Data is available, but not for further analysis. We attached database as requested only for this purpose but can not be publicly deposited. We believe that for the purpose of protecting the privacy of the research subjects, this data is not publicly available, and we do not have permission from participants or the IRB committee. In addition, there is at least a very small risk that some combination of the information could be used to deduce the identity of an individual. The collection of identifying information, if revealed, could harm the reputation of the research subjects. This information includes data about sexual attitudes and behavior, substance abuse, and other psychological information on the indigenous vulnerable population.
Once again, my apologies for the confusion, I am enclosing the database for your review. Because the type of file is not allowed here this was sent to the editorial office.